# Spectral cross-cumulants for multicolor super-resolved SOFI imaging

K. S. Grußmayer [1,2,5 ✉], S. Geissbuehler[2,5], A. Descloux [1,2], T. Lukes[1,2], M. Leutenegger[2,3], A. Radenovic [1] &
T. Lasser [2,4 ✉]

Super-resolution optical fluctuation imaging provides a resolution beyond the diffraction limit by analysing stochastic fluorescence fluctuations with higher-order statistics. Using $n^{th}$ order spatio-temporal cross-cumulants the spatial resolution and the sampling can be increased up to $n$-fold in all spatial dimensions. In this study, we extend the cumulant analysis into the spectral domain and propose a multicolor super-resolution scheme. The simultaneous acquisition of two spectral channels followed by spectral cross-cumulant analysis and unmixing increases the spectral sampling. The number of discriminable fluorophore species is thus not limited to the number of physical detection channels. Using two color channels, we demonstrate spectral unmixing of three fluorophore species in simulations and experiments in fixed and live cells. Based on an eigenvalue/vector analysis, we propose a scheme for an optimized spectral filter choice. Overall, our methodology provides a route for easy-to-implement multicolor sub-diffraction imaging using standard microscopes while conserving the spatial super-resolution property.

[1] Laboratory of Nanoscale Biology, École Polytechnique Fédérale de Lausanne, 1015 Lausanne, Switzerland. [2] Laboratoire d'Optique Biomédicale, École Polytechnique Fédérale de Lausanne, 1015 Lausanne, Switzerland. [3] Department of NanoBiophotonics, Max-Planck Institute for Biophysical Chemistry, Am Fassberg 11, 37077 Göttingen, Germany. [4] Max-Planck Institute for Polymer Research, Ackermannweg 10, 55128 Mainz, Germany. [5] These authors contributed equally: K. S. Grußmayer, S. Geissbuehler. ✉email: kristin.grussmayer@epfl.ch; theo.lasser@epfl.ch

Multicolor fluorescence microscopy is an invaluable tool for the study of cellular structures and function. Classical optical microscopes provide single-fluorophore sensitivity; however, the spatial resolution is limited due to diffraction[1]. During the last two decades, several super-resolution microscopy techniques overcame this limitation by exploiting on- and off-switching of fluorophores in an either deterministic or stochastic manner[2,3]. These methods often require fluorophores with high photostability (e.g. in stimulated emission depletion microscopy (STED)) or high off-to-on-switching ratios (e.g. in single-molecule localization microscopy (SMLM)). Super-resolution microscopes are only slowly finding their way into routine biological application due to complexity in instrument and sample preparation. Overcoming these hurdles with novel schemes may increase the adoption of advanced microscopy techniques. The abovementioned restrictions pose important road blocks for multicolor imaging applications. Several sub-diffraction imaging methods have demonstrated multicolor imaging[4,5]. They mainly rely on fluorophores with distinct spectra[6–9], more complex probes and labels for multiplexing[10–12], which are recorded sequentially in multiple channels, or on more complex approaches taking advantage of other fluorophore properties such as fluorescence lifetime[13].

The difficulty in obtaining optimal fluorophore behavior across the spectrum compatible with the constraints imposed by SMLM limits multicolor camera-based nanoscopy at present, i.e. it is problematic to identify suitable fluorophore multiplets. Workarounds such as spectrally resolved STORM[14] allow the use of several far-red-emitting fluorophores, albeit at the cost of much-increased hardware and analysis complexity. Thus, there is a need for robust and easy-to-implement multicolor sub-diffraction imaging.

Super-resolution optical fluctuation imaging (SOFI)[15,16] provides an elegant way of overcoming the diffraction limit in all spatial dimensions[17]. A classical widefield fluorescence microscope is used to acquire an image sequence of stochastically blinking fluorophores. Post-processing by calculating higher-order cumulants leads to a resolution improvement growing with the cumulant order. Unlike single-molecule localization-based techniques, SOFI does not require the spatio-temporal isolation of individual fluorophores' emissions[18,19] and is thus compatible with a wider range of blinking conditions and labeling densities. Thereby, SOFI simplifies the fluorophore selection, which is particularly welcome due to the inherent difficulty in labeling more than one protein in sufficient quality for super-resolution microscopy. Furthermore, due to the inherent optical sectioning properties of SOFI, the imaging of thick samples can be performed with widefield illumination and does not rely on physical background reduction, such as total internal reflection[20].

To date, cumulant analysis has been used for spatial super-resolution[15]. In spatio-temporal cross-cumulation[16], various combinations of neighboring pixels are used to obtain virtual pixels (i.e. a denser sampling of the super-resolved image). To our knowledge, only two-color SOFI has so far been demonstrated and used to visualize different structures in a cell and most experiments have been conducted sequentially[19,21–23]. However, by imaging multiple spectral channels step by step, correlations cannot be used, i.e. the cross-cumulation in between detection channels is not exploited.

In this work, we generalize the cumulant analysis by extending it into the spectral domain to pave the way towards a novel multicolor SOFI. Unlike other multicolor approaches, the cross-talk between the different physical color channels is a key ingredient for generating additional color channels. Here, we apply cross-cumulants between multiple simultaneously acquired spectral channels. The physical detection channels are thereby supplemented by virtual spectral channels to obtain a finer spectral sampling. The refined spectral sampling allows linear unmixing[24] of many distinct fluorophore colors with at least two recorded physical acquisition channels. We first introduce the theory behind spectral unmixing using spectral cross-cumulants and describe the workflow of our new multicolor imaging approach. We then demonstrate three-color imaging of different structures, fluorophores and filter sets in simulations and for fixed and live cells. We thus provide a route for easy-to-implement simultaneous multicolor sub-diffraction imaging with readily available microscopes.

## Results

**Spectral unmixing using spectral cross-cumulants.** Classical simultaneous multicolor imaging is achieved by adding dichroic filters in the imaging path to separate the light into two or more distinct spectral channels. For the following discussion, we will consider the simplest case where only two physical channels are used (see Supplementary Note 1 for a general discussion). For a given dichroic leading to an overall transmission $D_T(\lambda)$ and reflection $D_R(\lambda)$ spectrum) and fluorophore species with an emission spectrum $S_i(\lambda)$, we can define the transmission $T_i = \frac{\int_0^\infty S_i(\lambda)D_T(\lambda)d\lambda}{\int_0^\infty S_i(\lambda)d\lambda}$ and reflection coefficient $R_i = \frac{\int_0^\infty S_i(\lambda)D_R(\lambda)d\lambda}{\int_0^\infty S_i(\lambda)d\lambda}$ describing the collection of fluorescence signal which is detected according to the spectral response of the respective channel (for determination of $T_i$ and $R_i$ see Supplementary Note 2). If absorption and scattering of the dichroic is neglected, the relation $T_i = 1 - R_i$ holds. If we consider three spectrally distinct fluorophore species to be detected by two spectral detection channels, we can express the intensities $I_{R;T}(\boldsymbol{r})$ recorded in both spectral channels as

$$I_R(\boldsymbol{r}) = \sum_{i=1}^{3} R_i I_i(\boldsymbol{r})$$
$$I_T(\boldsymbol{r}) = \sum_{i=1}^{3} T_i I_i(\boldsymbol{r}) \quad , \tag{1}$$

with $I_i(\boldsymbol{r})$ being the intensity distribution of dye species $i$ measured at the detector pixel $\boldsymbol{r}$. This linear system can only be solved when using additional information to retrieve the images of the distinct dye species[24], as there are three unknowns $I_1(\boldsymbol{r})$, $I_2(\boldsymbol{r})$ and $I_3(\boldsymbol{r})$ for only two measurements $I_R(\boldsymbol{r})$ and $I_T(\boldsymbol{r})$.

Assuming stochastic, independent blinking of all fluorescent emitters, we can apply cumulant analysis on the time series $I_{R;T}(\boldsymbol{r}, t)$ recorded in the two physical channels and generate a so-called virtual spectral channel by computing the second-order cross-cumulants (Fig. 1). This virtual channel contains only crosstalk contributions from emitters that are detected and most important are correlated in both physical channels. Due to the additivity[15], the cumulant of multiple independent fluorophore species corresponds to the sum of the cumulants of each individual species and we can rewrite:

$$\begin{pmatrix} \kappa_{2,RR}(\boldsymbol{r}) \\ \kappa_{2,TR}(\boldsymbol{r}) \\ \kappa_{2,TT}(\boldsymbol{r}) \end{pmatrix} = \begin{pmatrix} R_1^2 & R_2^2 & R_3^2 \\ T_1 R_1 & T_2 R_2 & T_3 R_3 \\ T_1^2 & T_2^2 & T_3^2 \end{pmatrix} \begin{pmatrix} \kappa_2\{I_1(\boldsymbol{r}, t)\} \\ \kappa_2\{I_2(\boldsymbol{r}, t)\} \\ \kappa_2\{I_3(\boldsymbol{r}, t)\} \end{pmatrix}, \tag{2}$$

where $\kappa_{2,RR}$ and $\kappa_{2,TT}$ are second-order SOFI images calculated using the intensities of two-pixel combinations from the reflection and transmission channel[16], respectively. $\kappa_{2,TR}$ is the second-order spectral cross-cumulant image calculated using one pixel from each physical color channel. $\kappa_2\{I_i(\boldsymbol{r}, t)\} = \kappa_{2;i}$ denotes the second-order

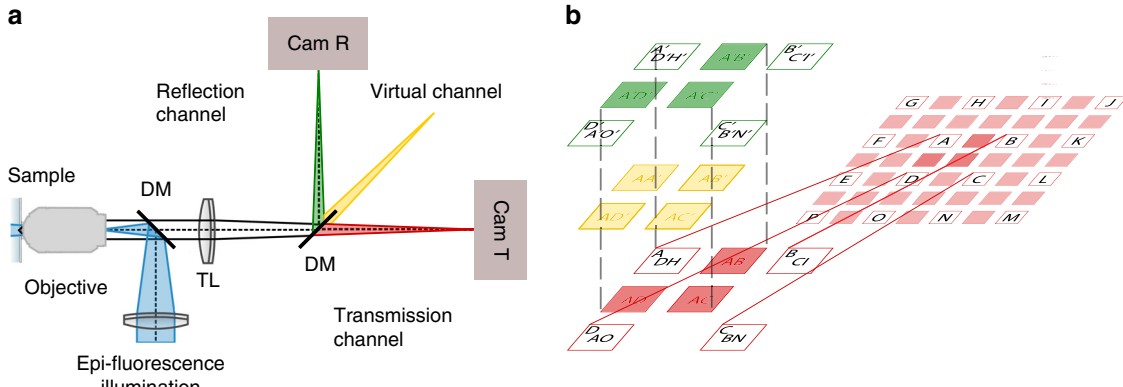

**Fig. 1 Cross-cumulant analysis between spectral channels. a** Simplified detection scheme with two physical spectral channels directed on two separate cameras (Cam R and Cam T, green and red). Spectral cross-cumulant analysis allows the generation of additional virtual spectral channels (yellow). DM dichroic mirror, TL tube lens, R reflection, T transmission. **b** Pixel combinations for the second-order cross-cumulant calculation. The cumulant analysis of each spectral channel (spectral auto-cumulant) is performed as described previously[16]. By cross-correlating intensities from different spectral channels (red and green) analogous to the computation of 'virtual' planes in multi-plane 3D SOFI[17], the additional 'virtual' cross-cumulant channel is computed (yellow). Single letters denote the original pixel matrix whereas multiplets of letters symbolize cross-cumulant calculation from combinations of original pixel intensities.

cumulant of the different dye species i. The possibility to compute an additional channel is the key to enabling unmixing by inversion of the linear system of equations (Eq. 2). We can thus recover the individual second-order cumulants for the three fluorophore species provided the spectral sensitivities of the dyes allow inversion of the matrix in Eq. 2. We provide a guide for selecting the optimal combination of dyes and filter set based on an eigenvalue/vector analysis in the Supplementary Note 3. So far, we have considered the simplest case of two physical channels and second-order cross-cumulant analysis. However, this cross-spectral cumulant analysis can be in principle generalized to obtain additional virtual spectral channels as shown in the Supplementary Note 1. For an $n^{\text{th}}$-order cumulant analysis of $N_{\text{P}}$ physical channels a total of $N_{\text{c}}$ cumulant color channels can be generated, with

$$N_{\text{c}} = \prod_{i=2}^{N_{\text{P}}} \frac{n+i-1}{i-1} \tag{3}$$

the number of distinct $n$-tuple combinations of the $N_{\text{P}}$ physical channels without permutations.

**Workflow spectral cross-cumulant analysis for multicolor SOFI.** In our experiments, we implemented two physical detection channels using a dichroic filter that spectrally splits the fluorescence emission and dispatches the reflected and transmitted light on two synchronized sCMOS cameras (Fig. 1a). To demonstrate the feasibility of the described multicolor SOFI with spectral unmixing, we first generated simulated datasets (see Methods).

Supplementary Figure 2 provides an example of the three common organic fluorophores Alexa Fluor 488, Atto565 and Alexa Fluor 647. Their emission spectra are weighted with the spectral responses of the reflection and transmission channels obtained by a multi-band dichroic and emission filter and a dichroic mirror at $\lambda = 594\,\text{nm}$. We generated image sequences (reflection and a transmission channel) for patches arranged in a grid of randomly blinking fluorophores representing either one of the three spectra. The workflow of the multicolor SOFI analysis is outlined in Fig. 2. After data acquisition, co-registration of the physical color channels is performed (Fig. 2, step 1, for details about the co-registration see Supplementary Note 4). The individual patches cannot be distinguished in the widefield images.

In the second step, the second-order spectral cross-cumulants are calculated to generate the three spectral channels $\kappa_{2,\text{RR}}(\boldsymbol{r})$, $\kappa_{2,\text{TR}}(\boldsymbol{r})$ and $\kappa_{2,\text{TT}}(\boldsymbol{r})$ as described above. The second-order cross-cumulant analysis adds a virtual color channel and already leads to a better spatial resolution inherent to raw SOFI images[15]. The individual patches can be identified, but the dyes are not yet unmixed. If the spectral cross-cumulants are computed with different spatial shifts and/or temporal delays, the resulting image generally has an inhomogeneous weight distribution arising from the spatio-temporal decorrelation of the signal[16] (as can be seen in Fig. 2, steps 2–3). We use zero time-delays in order to be more flexible in the range of detectable temporal fluctuations, but use spatial cross-cumulants to increase the virtual pixel grid density[16]. It is important to note, that the different pixels involved in the spectral cross-cumulants in step 2 of the algorithm should have the same spatial shifts for a specific output pixel such that the matrix inversion for color unmixing following in step 3 (Fig. 2, step 3) is possible. The fourth step consists in correcting for these inhomogeneous pixel weights by applying a distance-factor correction individually for each color or by maintaining the same mean for all pixel sub-grids, as it is applied for single-color spatial cross-cumulant SOFI. This flattening procedure cancels the spatial decorrelation arising from the finite PSF size[16] (see Fig. 2, step 4 and Supporting Note 1 for details). The last step (Fig. 2, step 5) consists in linearizing the cumulant response to brightness[25] by deconvolving each separate dye species image using Lucy–Richardson deconvolution, then taking the $n^{\text{th}}$ root and finally by reconvolving with a physically reasonable PSF. The patches can now be distinguished according to the spectra of the fluorophores and the patch size correlates with the wavelength (blue < yellow < magenta). Our analysis shows that spectral cross-cumulant analysis followed by unmixing tolerates a large range of intensities and blinking behavior. We can thus generate multi-color images with all the advantages inherent to SOFI such as optical sectioning, elimination of uncorrelated background and increased spatial resolution[15]. We investigated the influence of different photophysical properties of the fluorophores such as very closely overlapping spectra, varying photostability, on-time ratio and brightness on the performance of our multicolor analysis (Supplementary Note 5). For commonly used triplets such as Alexa Fluor 488, Atto565 and Alexa Fluor 647 the crosstalk remaining after our analysis is very low with 5% or less (Supplementary Table 2). When the fluorophore emission

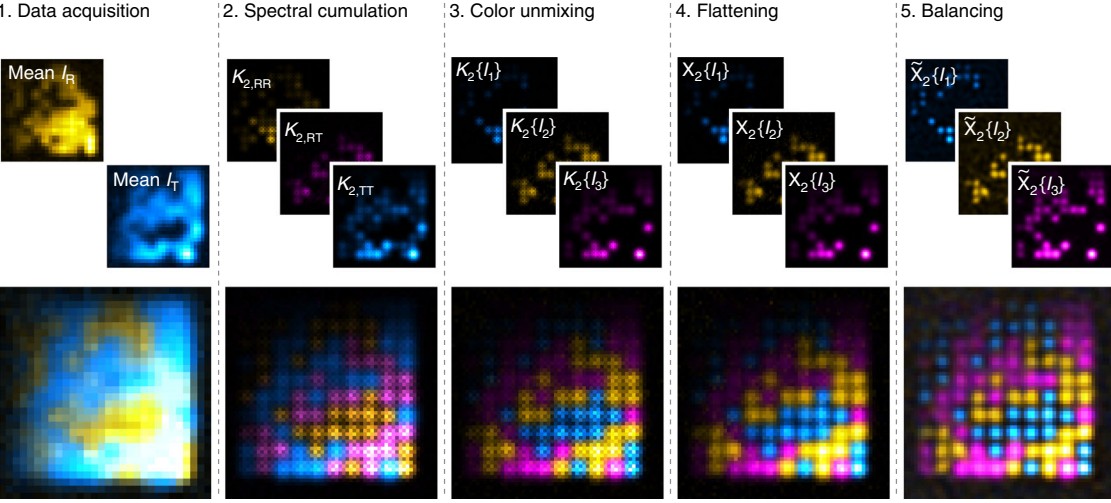

**Fig. 2 Workflow of multicolor SOFI imaging by spectral cross-cumulant analysis followed by linear unmixing using simulations.** A 2.5 pixel grid of patches with ~20 nm radius and 2 fluorophores each (~1300 emitter $\mu m^{-2}$) was simulated. Alexa488 (blue hot), Atto565 (yellow hot) or Alexa647 (magenta hot) spectral properties are randomly assigned to each patch. $I_{on}$ varies from top to bottom (200–1100 photons) and the on-ratio varies from left to right (0.01–0.1). 4000 frames with negligible photobleaching were analyzed.

maxima are only ~10 nm apart, e.g. for Abberior Flip 565 ($\lambda_{abs/em,max}$ = 566/580nm), Atto565 ($\lambda_{abs/em,max}$ = 564/590nm) and Alexa Fluor 568 ($\lambda_{abs/em,max}$ = 578/603nm), the remaining crosstalk is at worst 14% (Supplementary Table 3). Our exemplary analysis of filament-like structures shows that our multicolor processing is reliable for a wide range of expected fluorophore behavior.

### Experimental results

As a first experimental demonstration of multicolor SOFI with spectral unmixing, we imaged a time series of fixed HeLa cells stained with three different fluorophores. We collected the fluorescence light split across a dichroic at on two synchronized sCMOS cameras as mentioned above (Fig. 3a). The spectral response of the reflection and transmission channel as well as the weighted emission of the three fluorophores matches the values of the simulations shown above (Fig. 2 and Supplementary Fig. 1). We labeled microtubules with Alexa Fluor 488 via antibody staining, glycoproteins and sialic acid using wheat-germ agglutinin (WGA)-Atto565 and Lamin B1 in the nuclear membrane with Alexa Fluor 647 also via antibody staining. Appropriate blinking for SOFI processing was achieved using a buffer with thiols and oxygen scavengers; the fluorophores were excited with three different lasers at 488, 561 and 635 nm wavelength and moderate illumination intensities. We evaluated 2000 frames acquired with an exposure time of 20 ms per frame. The cumulant calculation was split into 10 subsequences to minimize the impact of photobleaching on SOFI analysis. Based on the conventional dual channel image (Fig. 3a), separation of the different labels is impossible. Spectral cross-correlation leads to the known optical sectioning and background reduction inherent to SOFI analysis[15] and generates the third virtual channel $\kappa_{2,RT}$ that is dominated by the WGA signal (Fig. 3b, blue and Supplementary Fig. 11). Supplementary Figure 11 illustrates the remaining spectral crosstalk present in the three channels. The final unmixed color channels (Fig. 3c–f) manage to separate the cell's microtubule network (Alexa Fluor 488, blue) from the diffuse WGA-staining (Atto565, green) and the typical Lamin B1 structure in the nuclear membrane (Alexa Fluor 647, red). WGA is a lectin that labels the cell membrane, but also the Golgi apparatus in the periphery of the nucleus and nuclear pore complex proteins in the nuclear membrane[26]. Residual crosstalk

and imperfectly reconstructed microtubules are only apparent in a small part of the image in the periphery of the nucleus (see arrow in Fig. 3f), where many microtubules overlap with extremely bright Golgi staining. Otherwise, the algorithm performs remarkably well although the image contains overlapping structures across the field of view and we use common fluorophores that have a high on-ratio under the applied moderate illumination conditions (Atto565 and Alexa Fluor 488). Analysis of the spatial frequency content of the three-color channels also confirms the increased resolution after multicolor SOFI processing (Supplementary Note 6).

As a second example, we imaged two densely labeled, almost completely overlapping structures in the nucleus of the cell. The DNA was stained with Hoechst-Janelia Fluor 549 with a blue-shifted spectrum compared to the previously used Atto565 and the nuclear membrane was visualized as before (antibody staining of Lamin B1 with Alexa Fluor 647), with the focus on the bottom of the nucleus. The third structure is again microtubules that are labeled with Alexa Fluor 488 via antibodies. One can appreciate that our spectral cross-cumulant unmixing approach manages to disentangle the two nuclear stains due to the specific spatio-temporal fluctuations of the different fluorophores. Distinctly different small structures are revealed showing the expected patterns such as small folds in the nuclear membrane (Supplementary Figs. 12 and 13), whereas the overall shape appears in both the Janelia Fluor 549 and the Alexa Fluor 647 channel. Additional experiments confirm that spectral cross-cumulant analysis can be applied using different dichroic beam splitters (Supplementary Figs. 14 and 15) and for fluorophores having a larger spectral overlap (Supplementary Figs. 16 and 17). Only two lasers were needed for the excitation of the three species (Alexa Fluor 568 secondary antibody staining of mitochondria, Hoechst-Janelia Fluor 549 labeling and Alexa Fluor 647 microtubule staining), further reducing the complexity of the experimental setup.

Next, we performed three-color live-cell imaging where the simultaneous data acquisition of our multicolor approach is a key advantage. We used a reversibly photoswitchable fluorescent protein fusion construct that we have applied previously for SOFI[17] (Vimentin–Dreiklang) and two live-cell compatible small molecule stains to label mitochondria (Mitotracker Deep Red FM) and wheat-germ agglutinin (WGA)-AbberiorFlip565. The

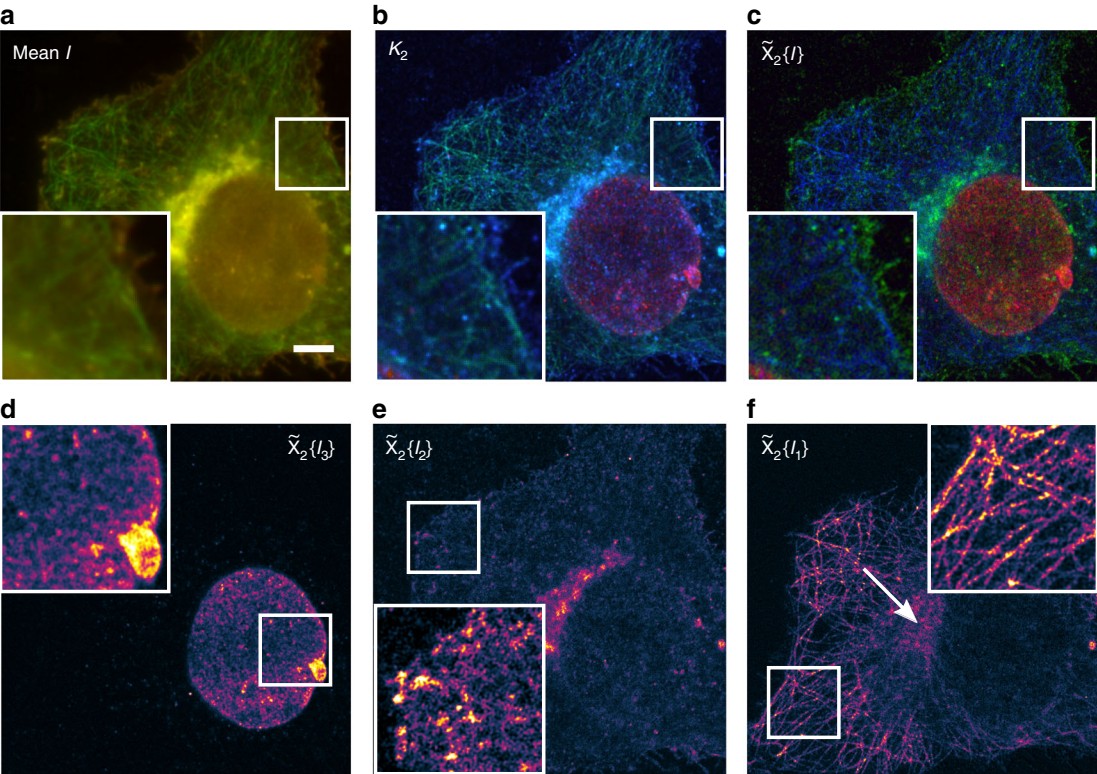

**Fig. 3 Multicolor SOFI of the cytoskeleton, nucleus and cellular membranes of HeLa cells. a** Overlay of the average intensity acquired in the reflection (green) and transmission (red) channel using 200 mM MEA with oxygen scavenging and about 0.5 kW cm$^{-2}$ 488 nm, 1.25 kW cm$^{-2}$ 561 nm and 1.3 kW cm$^{-2}$ 635 nm illumination intensity; 2000 frames and 20 ms exposure time. **b** RGB composite image of the second-order spectral cross-cumulant images with $\kappa_{2,RR}$(green), $\kappa_{2,RT}$(blue) and $\kappa_{2,TT}$(red). **c** RGB composite image of the unmixed, flattened and deconvolved SOFI images with **d** Alexa Fluor 647 secondary antibody stained nuclear membrane (red). **e** Wheat-germ agglutinin-Atto565 labeling (green, $\widetilde{\chi}_2\{I_2\}$) and **f** Alexa Fluor 488 secondary antibody stained microtubules (blue, $\widetilde{\chi}_2\{I_1\}$). The separate unmixed images are displayed using the morgenstemning colormap[27]. Scale bar 5 μm and insets 8.64 μm x 8.64 μm. This figure is representative of more than 10 multicolor SOFI reconstructions obtained from at least two independent experiments.

appropriate fluorophore blinking was achieved using 488 nm light for imaging and off-switching of Dreiklang, with on-switching via thermal relaxation; Mitotracker was imaged and switched off by 635 nm excitation and the self-blinking spiroamide dye conjugate WGA-AbberiorFlip565 was imaged using 561 nm excitation. This proof-of-principle multicolor experiment shows the cytoskeleton elements Vimentin, mitochondria and WGA (presumably WGA accumulated in internalized vesicles or membrane folds as we are focusing just above the basal membrane). The three-color unmixed images are displayed in Fig. 4b and Supplementary Fig. 18. Fig. 4c illustrates the multicolor workflow for a zoomed in region in our live-cell example. The initial images in the reflection channel $I_G$ contain mostly signal from the Vimentin–Dreiklang fluorescent protein construct mixed with sparse WGA agglomeration (see arrows in Fig. 4c for WGA overlaying Vimentin filaments). The spectral cross-cumulant image $\kappa_{2,RR}$ reflects the blinking signals and the spectral unmixing finally removes the WGA-AbberiorFlip565 signal from the Dreiklang channel $\widetilde{\chi}_2\{I_1\}$ to uncover previously hidden Vimentin structure (see line profiles in Fig. 4c) and note the background reduction. Supplementary Figure 19 shows the corresponding images for all channels.

Live-cell super-resolution imaging is challenging, considering the intricate relationship between phototoxicity, imaging speed, resolution increase, spatial sampling and possible motion blur[28]. For biological investigations, these parameters need to be carefully considered for each target and fluorophore. The directional movement of fluorophores during the imaging limits the useful spatio-temporal window were correlations can be extracted,

which in turn lowers the SOFI signal- to- noise ratio and the resolution. In general, the resolution is compromised when the frame acquisition time and the feature velocity lead to a displacement on the order of the attainable resolution[29]. Diffusion of fluorophores for an overall stationary structure has only minor effects on the SOFI signal while substantially improving the spatial sampling[30]. In general, SOFI tolerates higher labeling densities, on-time ratios, lower signal-to-noise ratio and needs less frames to reconstruct an image than SMLM. There have been select applications, where single-color SMLM achieved <1 s time resolution, but usually tens of seconds acquisitions are necessary[28,31]. We previously demonstrated live-cell SOFI with acquisition times ~1 s (~325 frames[17]) using optimized, fast switching of Dreiklang and others achieved live-cell SOFI for several fluorescent proteins by analyzing ~500 frames[32].

## Discussion

In this work, we have shown that the spectral sampling can be refined based on a spectral cross-cumulant calculation between simultaneously acquired color channels. The simultaneous acquisition of multiple color channels and spectral cross-cumulation allows to unmix several fluorophore species even with strongly overlapping emission spectra, where the number of species is not limited to the number of physical spectral channels. Using a basic two-color detection scheme, we validate the spectral unmixing of three fluorophore species labeling in simulations and experiments. We demonstrate multicolor imaging of different

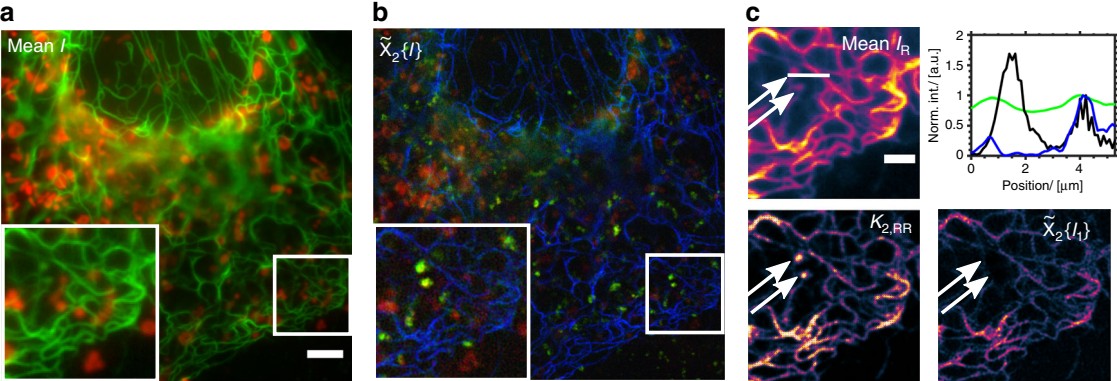

**Fig. 4 Multicolor SOFI live-cell imaging of the cytoskeleton, mitochondria and wheat-germ agglutinin in COS-7 cells. a** Overlay of the average intensity acquired in the reflection (green) and transmission (red) channel of live cells in Hanks balanced salt solution and about 0.8 kW cm$^{-2}$ 488 nm, 0.8 W cm$^{-2}$ 561 nm and <1 kW cm$^{-2}$ 635 nm illumination intensity; 1000 frames and 20 ms exposure time. Scale bar 5 μm. **b** RGB composite image of the unmixed, flattened and deconvolved SOFI images with Mitotracker Deep Red FM stained mitochondria (red, $\widetilde{\chi}_2\{I_3\}$), accumulated wheat-germ agglutinin-AbberiorFlip565 labeling (green, $\widetilde{\chi}_2\{I_2\}$) and Vimentin–Dreiklang fluorescent protein expression (blue, $\widetilde{\chi}_2\{I_1\}$). **c** Close-up of the ROI indicated in **a** and **b** showing the mean intensity in the reflection channel $I_R$, the second-order spectral cross-cumulant image $\kappa_{2,RR}$ and the unmixed image in the Dreiklang channel $\widetilde{\chi}_2\{I_1\}$, all displayed using the morgenstemning colormap[27] and comparison of the normalized intensity profiles along the indicated line (green $I_R$, black $\kappa_{2,RR}$, blue $\widetilde{\chi}_2\{I_1\}$). Scale bar 2 μm. This figure is representative of more than three multicolor SOFI reconstructions.

structures, fluorophores and filter sets in fixed and live HeLa and COS-7 cells. To our knowledge, this is the first published demonstration of three-color SOFI revealing three subcellular components. We also provide a guideline for optimized spectral filter choice based on an eigenvalue/ vector analysis. The proposed strategy corresponds to the experimentally most straightforward implementation, nowadays frequently met in commercial setups via image splitting units using one camera. The possibility of cross-cumulating between color channels thus translates the concept of spatial super-resolution to spectral super-sampling. As we formulated multicolor spectral cross-cumulant SOFI in the theoretical framework originally devised for spatially super-resolved SOFI, it is intrinsically compatible with previous developments such as 3D SOFI[17] and bSOFI[25]. In particular, our approach could be combined with the recently published multiple-tau (mt)-pcSOFI[33] to double the number of imaged structures, provided suitable pairs of fluorophores (organic dyes, fluorescent proteins, polymer dots, etc) are available. mt-pcSOFI achieves multiplexing using a single-color channel by exploiting differences in blinking kinetics of dyes when calculating SOFI cumulants.

Spectral cross-cumulant multicolor SOFI makes use of the crosstalk between adjacent spectral channels. The crosstalk is key in extracting the third color and is not a limitation or artifact like in classical multicolor imaging. This concept exploits the intrinsic properties of cumulant analysis in the spectral domain while maintaining the super-resolution performance in the spatial domain. Alternative super-resolution concepts based on structured illumination or targeted switching[28] require an instrument modification adapted for each spectral channel. Super-resolution approaches relying on the stochastic fluorophore switching generally require simpler instrumentation. Our approach does not need any hardware modification and can be easily adopted in ubiquitous 2-channel widefield microscopes. Compared to other super-resolution multicolor approaches, we used standard labeling with organic dyes and fluorescent proteins compatible with live-cell imaging. There is no need for sophisticated probes that are difficult to obtain and use (e.g. DNA probes for exchange-PAINT[10] or activator-reporter pairs in STORM[12]). Our analysis preserves all the advantages inherent to SOFI such as optical sectioning, elimination of uncorrelated background and increased spatial resolution[15]. In particular, cumulant analysis tolerates

high fluorophore labeling densities, high on-time ratios and low signal-to-noise ratios[18,19]. This facilitates fluorophore selection and experimental realization (e.g. no need for high power lasers), which is especially important for multicolor imaging. Spectral cross-cumulation thus enables multicolor super-resolved imaging when blinking conditions for single-molecule localization microscopy are hard or impossible to achieve[18,34,35]. The demonstrated 2$^{nd}$ order analysis offers 3-color imaging with up to twofold resolution increase. In principle, our approach is not limited to three colors; e.g. 4-color imaging with 2 spectral channels could be achieved through 3$^{rd}$ order SOFI at up to 3-fold higher resolution (see theory in the Supplementary Information). Obviously, simultaneous data acquisition saves time compared to sequential imaging and SOFI generally achieves super-resolution using less frames and shorter acquisition times as well as lower laser intensities than SMLM[18,19]. This is important for live-cell imaging to minimize motion blur and phototoxicity.

To conclude, we presented a generalized cumulant concept for multicolor spectral cross-cumulant SOFI analysis with thorough proof-of principle simulations and experiments in live and fixed cells. This approach proved to be robust and compatible with a large range of fluorophores enabling sub-diffraction imaging of several cellular structures while using readily available microscope hardware.

## Methods

**Simulations.** The code for simulations was based on the previously published "SOFI Simulation Tool" software package that simulated images of fluorophores from a single species recorded on one camera[36]. Briefly, for each fluorophore species, single emitters are randomly placed according to a certain spatial density and spatial distributions. For each emitter, the blinking behavior is modeled as a time-continuous Markovian process with exponential probability distribution functions with average blinking on-time $\tau_{on}$ and off-time $\tau_{off}$. The camera detects on average $I_{on}$ photons in the on-state. Fluorophore photobleaching is also considered by a single exponential decay with average bleaching time $\tau_{pb}$. The PSF is assumed to be a rotationally symmetric 2D Gaussian with a standard deviation according to the numerical aperture, camera pixel size and wavelength of the fluorophore. A spatially constant background $I_b$ is added to the total fluorescence signal. The summed signal per frame is then split in the transmission and reflection channel according to the emission spectra of the fluorophore and the spectral response curve of the microscope. The generated image stacks $I_{R,i}(\mathbf{r}, t)$ and $I_{T,i}(\mathbf{r}, t)$ for the different fluorophore species $i$ are then summed up and each pixel intensity is subjected to Poissonian noise. The intensity per pixel is converted to electric charge according to the quantum efficiency and gain of the camera and Gaussian

noise with a standard deviation related to dark noise is added to obtain the final time series $I_R(r, t)$ and $I_T(r, t)$.

**Chemicals**. Unless noted otherwise, all chemicals were purchased at Sigma-Aldrich.

**Cell culture**. HeLa cells and COS-7 cells were cultured at 37 °C and 5% $CO_2$ using DMEM high glucose with pyruvate (4.5 g l$^{-1}$ glucose, with GlutaMAX$^{TM}$ supplement) supplemented with 10% fetal bovine serum and 1 × penicillin–streptomycin (all gibco®, ThermoFisher Scientific) or DMEM high glucose w/o phenol red (4.5 g l$^{-1}$ glucose) supplemented with 4 mM L-glutamine, 10% fetal bovine serum and 1 × penicillin–streptomycin (all gibco®, ThermoFisher Scientific). HeLa cells were from ATCC (ATCC® CCL-2™) and COS-7 cells were a kind gift of the Manley lab (LEB, EPFL).

Cells were seeded in Lab-tek® II chambered cover slides (nunc) 1–2 days before fixation in DMEM high glucose w/o phenol red (4.5 g l$^{-1}$ glucose) supplemented with 4 mM L-gluthamine, 10% fetal bovine serum and 1 × penicillin–streptomycin (all gibco®, ThermoFisher Scientific).

**HeLa cell fixation and staining**. HeLa cells were washed twice in pre-warmed buffer (microtubule stabilizing buffer (MTSB)): 100 mM PIPES pH 6.8, 2 mM $MgCl_2$, 5 mM EGTA or PBS for wheat-germ agglutinin (WGA) staining), followed by application of pre-warmed fixation buffer (3.7% paraformaldehyde (PFA), 0.2% Triton X-100 in MTSB or 3.7% paraformaldehyde (PFA) in PBS for wheat-germ agglutinin (WGA) staining) for 15 min at room temperature (RT). Cells were then washed three times for 5 min each with 1 × PBS and stored in 50% glycerol in 1 × PBS at 4 °C or the immunostaining protocol was continued to prepare samples for fluorescence imaging.

Fixed and permeabilized cells were blocked with 3% BSA in 1 × PBS and 0.05% Triton X-100 for 60 min at RT or overnight at 4 °C.

WGA-staining: Cells that were fixed without permeabilization were stained with 5 µg ml$^{-1}$ WGA-Atto565 (preparation see below) for 10 min followed by three times 5 min washes with 1 × PBS. Subsequently, the cells were blocked using blocking buffer containing 0.2% Triton X-100.

Microtubule and nuclear envelope staining: The blocked samples were immediately incubated with a mix of primary anti-tubulin antibody (1 mg ml$^{-1}$ DM1a mouse monoclonal (ab80779) 1:150 dilution, Abcam) and anti-Lamin B1 antibody (1 mg ml$^{-1}$ rabbit polyclonal (ab16048) 1:400 dilution, Abcam) in antibody incubation buffer for 60 min at RT (AIB: 1% BSA in 1 × PBS and 0.05% Triton X-100). Cells were then washed three times for 5 min each with AIB, followed by incubation with a mix of donkey anti-mouse-Alexa Fluor 647 antibody (0.005 mg ml$^{-1}$ Invitrogen) and donkey anti-rabbit-Alexa Fluor 488 antibody (0.01 mg ml$^{-1}$ Invitrogen) for 60 min at RT. This and all subsequent steps were performed in the dark. Cells were again washed three times for 5 min each with AIB, optionally subjected to DNA staining and incubated for 15 min post-fixation with 2% PFA in 1 × PBS followed by three 5 min washes with PBS. Cells were imaged immediately or stored in 50% glycerol in 1 × PBS at 4 °C until imaging.

DNA staining: 10 µM Hoechst-Janelia Fluor 549 in PBS was incubated for 10 min followed by three times 5 min washes with 1 × PBS.

**COS-7 cells fixation and staining**. The protocol is similar as described previously by Chazeau et al.[37]. Cells were washed twice in pre-warmed DMEM w/o phenol red (see cell culture) following 90 s incubation with extraction buffer (microtubule stabilizing buffer 2 (MTSB2: 80 mM PIPES, 7 mM $MgCl_2$, 1 mM EGTA, 150 mM NaCl, 5 mM D-glucose adjust pH to 6.8 using KOH)) with freshly added 0.3% Triton X-100 (AppliChem) and 0.25% glutaraldehyde (stock solution 50% electron microscopy grade, Electron Microscopy Sciences). Immediately afterwards, pre-warmed 4% paraformaldehyde (PFA) in PBS was incubated for 10 min at room temperature (RT). Cells were then washed three times for 5 min each with 1 × PBS and stored in 50% glycerol in 1 × PBS at 4 °C or the immunostaining protocol was continued. Next, a freshly prepared solution of 10 mM $NaBH_4$ in 1 × PBS was incubated on the cells for 7 min followed by one quick wash in 1 × PBS, and two washes 10 min 1 × PBS on an orbital shaker. Cells were permeabilized in PBS with 0.25% Triton X-100 for 7 min followed by blocking with blocking buffer (BB: 2% (w/v) BSA, 10 mM glycine, 50 mM ammonium chloride $NH_4Cl$ in PBS pH 7.4 for 60 min at RT or overnight at 4 °C).

The blocked samples were incubated with primary anti-tubulin antibody (clone B-5-1-2 ascites fluid 1:200 dilution, Sigma-Aldrich) and primary anti-TOMM20 antibody ([EPR15581], 1:50 dilution, Abcam) in BB for 60 min at RT. Cells were then washed three times for 5 min each with BB, followed by incubation with donkey anti-mouse-Alexa Fluor 647 antibody (donkey anti-mouse (H + L) highly cross-adsorbed at 0.005 mg ml$^{-1}$ Invitrogen) and donkey anti-rabbit-Alexa Fluor 568 (donkey anti-mouse (H + L) highly cross-adsorbed at 0.005 mg ml$^{-1}$ Invitrogen) for 60 min at RT. This and all subsequent steps were performed in the dark. Cells were again washed three times for 5 min each with BB and incubated for 10 min post-fixation with 2% PFA in 1 × PBS followed by three 5 min washes with PBS. Cells were imaged immediately or stored in 50% glycerol in 1 × PBS at 4 °C until SOFI imaging. Just before imaging, 2 µM Hoechst-Janelia Fluor 549 in PBS was incubated for 10 min followed by three times 5 min washes with 1 × PBS.

**COS-7 cells protein expression, staining and live-cell imaging**. A total of 150,000 cells were seeded on 25 mm high-precision No. 1.5 borosilicate coverslips (Marienfeld) in 6 well plates (ThermoFisher Scientific) 1 day before transfection in DMEM w/o phenol red (see cell culture). Each chamber was transfected with 2 µg of the plasmid pMD-Vim-Dreiklang[17] using Lipofectamine 3000 (ThermoFisher Scientific) according to manufacturer's instructions and imaged about 24 h post transfection. Before imaging, the coverslip was incubated with 600 nM Mitotracker Deep Red FM in DMEM w/o phenol red for ~30 min at 37 °C and 5% $CO_2$, followed by 10 min incubation of ~10 µg ml$^{-1}$ WGA-AbberiorFlip565 in Hanks balanced salt solution w/$MgCl_2$ and $CaCl_2$ (HBSS, Gibco). After 1× washing in HBSS the samples were imaged in HBSS at RT for up to 30 min. For imaging, we preswitch Dreiklang using 488 nm light for 10 s before starting data acquisition.

**Preparation of labeled proteins**. 2 mg ml$^{-1}$ WGA (Vector Labs) was incubated for 1 h at RT while shaking with Atto565-NHS ester (Atto-tec) or AbberiorFlip565-NHS ester (Abberior) at a ratio of 1:6 with the pH raised to 8.3 using sodium bicarbonate. The mixture was purified using illustra NAP Columns (GE Healthcare) according to manufacturer's instructions and eluted with slightly acidic PBS to recover labeled antibody at neutral pH. The protein concentration was estimated by absorption spectrometry to 0.5 mg ml$^{-1}$ WGA-Atto565 and to ~1 mg ml$^{-1}$ WGA-AbberiorFlip565.

**Imaging buffer for fixed cells**. The samples were imagined in a 50 mM Tris-HCl pH 8.0, 10 mM NaCl buffer containing an enzymatic oxygen scavenging system (2.5 mM protocatechuic acid (PCA) and 50 nM Protocatechuate-3,4-Dioxygenase from *Pseudomonas* sp. (PCD) with >3 U mg$^{-1}$) and a thiol (2-Mercaptoethylamine). The thiol and a stock solution of 100 mM PCA in water, pH adjusted to 9.0 with NaOH, were always prepared fresh. PCD was aliquoted at a concentration of 10 µM in storage buffer (100 mM Tris-HCl pH 8.0, 50% glycerol, 50 mM KCl, 1 mM EDTA) at −20 °C and thawed immediately before use.

**Microscope setup**. All imaging was performed with a custom built widefield fluorescence microscope equipped with a 200 mW 405 nm laser (MLL-III-405-200mW), a 1 W 635 nm laser (SD-635-HS-1W, both Roithner Lasertechnik), a 350 mW 561 nm laser (Gem561, Laser Quantum) and a 200 mW 488 nm laser (iBEAM-SMART-488-S-HP, Toptica Photonics) and individual shutters controlled via an arduino in front of each laser. The lasers were combined and focused into the back focal plane of a Nikon SR Plan Apo IR 60 × 1.27 NA WI objective. The fluorescence light was filtered using a combination of a dichroic mirror and a multi-band emission filter (Quad Line Beamsplitter R405/488/561/635 flat and Quad Line Laser Rejectionband ZET405/488/561/640, both AHF Analysetechnik).

In the detection path, the light is focused by a 200 nm tube lens before being split by an emission dichroic (Laser Beamsplitter zt 594 RDC or Beamsplitter HC BS 640 imaging, both AHF Analysentechnik) and directed on two synchronized sCMOS cameras (ORCA Flash 4.0, Hamamatsu; back projected pixel size of 108 nm). For translating the sample, the microscope is equipped with an xy motorized stage (SCAN$^{plus}$ IM 120 × 80 Maerzhaeuser with Tango Desktop driver). Focus stabilization is provided by a nanometer z positioning stage (Nano-ZL300-M; Mad City Labs with Nano-Drive C controller) driven by an optical feedback system similar to ref. [38]. Data acquisition were performed using MicroManager 1.4 and a custom beanshell script controlling the laser shutters, camera acquisition and data saving.

**Data processing**. The spectral cross-cumulant multicolor algorithm was implemented in MATLAB R2016b (Mathworks). We adapted and extended the bSOFI MATLAB package used in ref. [25]. For resolution estimation, we used the Fiji plugin of image decorrelation analysis[39].

**Reporting summary**. Further information on research design is available in the Nature Research Reporting Summary linked to this article.

## Data availability
All data needed to evaluate the conclusions in the paper are present in the paper and/or the Supplementary Information. All relevant raw data are available from the authors upon reasonable request.

## Code availability
The software package, including a simulation dataset and instructions, is available under the GNU general public license. The instructions can also be found on https://www.epfl.ch/labs/lben/sofi-packages/ and the software package can be downloaded at https://www.epfl.ch/labs/lben/wp-content/uploads/2020/05/multicolor_sofi_v2.3.zip.

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

## Acknowledgements

Hoechst-Janelia Fluor 549 was a kind gift from Luke Laevis (Janelia Research Campus, Howard Hughes Medical Institute, USA) and COS-7 cells were a kind gift of the Manley lab (EPFL). We gratefully acknowledge the support of NVIDIA Corporation with the donation of the Titan Xp GPU used for this research. K.S.G. has received funding from the European Union's Horizon 2020 research and innovation program under the Marie Skłodowska-Curie Grant Agreement No. [750528]. M.L. thanks Prof. Stefan W. Hell for the research position in his department.

## Author contributions

K.S.G. performed experiments and simulations. S.G. and M.L. established the multicolor spectral cross-cumulant concept and wrote the initial analysis software. A.D., T.L. and K.S.G. adapted and extended the analysis software and A.D. and K.S.G. analyzed the data. T.L. contributed to the theory of the multicolor spectral cross-cumulant concept and formulated the eigenvalue/eigenvector analysis with A.D. and K.G.. T.L. and A.R. supervised the project. K.S.G wrote the manuscript with input from all authors.

## Competing interests

The authors declare no competing interests.
