## [Peer Review File · Nature Communications]

Reviewers' Comments:

Reviewer #1:

Remarks to the Author:

The SOFI technique, like all fluorescence SR methods, makes use of the independence of individual fluorophores' fluorescent state. It is certainly possible, and has been demonstrated, that a two-channel system can be used to classify several different isolated fluorophores by their ratio in the two channels. What is described here is the analogous approach under the framework of the SOFI method. Although the mathematics might seem somewhat involved, this is the simplest and most straightforward approach to take. Still, it is a clever idea that is also practically useful for those that are using the SOFI method because many TIRF or widefield systems have a two-channel splitter system (they are cheaper and more ubiquitous commercially, and easier to custom build) but much fewer would have three or more channels.

I enjoyed reading the manuscript. It is well written with the method laid out in a straightforward manner. Every question that occurred to me as I started reading was addressed somewhere in the manuscript or SI. I did simulate data and verify that eq 2 is correct and works. The following analyses after eq 2 are not particular to this paper and have been demonstrated previously. The SI explores the performance under realistic situations.

Reviewer #2:

Remarks to the Author:

This paper described a method to do 3 color super-resolution SOFI images using two physical detection channels. It generalized a previously described spatial-temporal cross-cumulant approach between pixels on the same camera, to cross-cumulant between spectrally different channels. The authors demonstrated the principle using simulation and illustrated the application with fixed sample imaging. The theory behind the approach was clearly explained and application was sufficient to demonstrate the point of color unmixing. However, the reviewer has concerns about the novelty and proper benchmarking.

1. The concept of cross-cumulant is not that new and has been applied previously by the authors and other group. The value provided in the paper is a generalization to multi-color. If the authors provided software packages that help users performing such analysis, it would be a plus.
2. There are many existing super-resolution methods capable of doing what the current method can do, for example, structural illumination, DNA-paint, etc. The 3 color imaging can be done even with sequential SOFI with 3 different colors. The simultaneous 2-channel acquisition can save some time, but for fixed sample, this advantage is marginal. So the authors should really provide some applications that illustrate the unique advantages that other methods cannot do or this method can do much better.

Minor

1. In Figure 3, the authors could add some analysis to the performance of the method, rather than just the figure and the zoom.
2. Figure 3 and 4 provided similar points of color unmixing and did not add much.

We are thankful to the reviewers for evaluating our manuscript. We analyzed in detail their remarks and addressed them point-by-point as described below.

All the changes, additional information and Figures in the revised version of the manuscript and supplementary information are highlighted in yellow.

Reviewer #1 (Remarks to the Author):

The SOFI technique, like all fluorescence SR methods, makes use of the independence of individual fluorophores' fluorescent state. It is certainly possible, and has been demonstrated, that a two-channel system can be used to classify several different isolated fluorophores by their ratio in the two channels. What is described here is the analogous approach under the framework of the SOFI method. Although the mathematics might seem somewhat involved, this is the simplest and most straightforward approach to take. Still, it is a clever idea that is also practically useful for those that are using the SOFI method because many TIRF or widefield systems have a two-channel splitter system (they are cheaper and more ubiquitous commercially, and easier to custom build) but much fewer would have three or more channels.

I enjoyed reading the manuscript. It is well written with the method laid out in a straightforward manner. Every question that occurred to me as I started reading was addressed somewhere in the manuscript or SI. I did simulate data and verify that eq 2 is correct and works. The following analyses after eq 2 are not particular to this paper and have been demonstrated previously. The SI explores the performance under realistic situations.

Reply: We thank the reviewer for the positive comments and the appreciation of our work. We have now included our software package with a simulated test dataset and instructions to facilitate implementation of our method. We provide the software package to the editor and reviewers and it will be deposited in an open source repository upon publication.

Reviewer #2 (Remarks to the Author):

This paper described a method to do 3 color super-resolution SOFI images using two physical detection channels. It generalized a previously described spatial-temporal cross-cumulant approach between pixels on the same camera, to cross-cumulant between spectrally different channels. The authors demonstrated the principle using simulation and illustrated the application with fixed sample imaging. The theory behind the approach was clearly explained and application was sufficient to demonstrate the point of color unmixing. However, the reviewer has concerns about the novelty and proper benchmarking.

Reply: We thank the reviewer for his appreciation of our multicolor spectral cross-cumulant theory section and the successful demonstration of the approach.

Reviewer #2:

1. The concept of cross-cumulant is not that new and has been applied previously by the authors and

other group. The value provided in the paper is a generalization to multi-color. If the authors provided software packages that help users performing such analysis, it would be a plus.

Reply: We agree with the reviewer that providing our software will help users perform the multicolor analysis. We included our software package with a simulation dataset and instructions to facilitate the implementation of our method. We provide the software package to the editor and reviewers and it will be deposited in an open source repository upon publication.

Reviewer #2:

2. There are many existing super-resolution methods capable of doing what the current method can do, for example, structural illumination, DNA-paint, etc. The 3 color imaging can be done even with sequential SOFI with 3 different colors. The simultaneous 2-channel acquisition can save some time, but for fixed sample, this advantage is marginal. So the authors should really provide some applications that illustrate the unique advantages that other methods cannot do or this method can do much better.

Reply: The reviewer is correct, there are several super-resolution methods providing color imaging, as mentioned in our introduction. However, to our best knowledge, we are not aware of any publication demonstrating (not even sequential) SOFI with 3 different colors labeling three different structures. Our study thus demonstrates the first three color SOFI (we previously mentioned this in the introduction line 71 and now added it in the Summary and Discussion section).

In addition, there are substantial advantages of our proposed method as:

- Our approach does not need any hardware modification and can be easily adopted in existing widefield microscopes with two color channels, e.g. using a commercially ubiquitous two-channel splitter system (compared to e.g. SIM).
- We used standard labeling approaches with organic dyes and fluorescent proteins and there is no need for sophisticated probes that are more difficult to obtain and use (compared to e.g. DNA-PAINT or activator-reporter pairs in STORM).
- Our method allows unmixing of dyes with highly overlapping spectra (see simulations in SI).
- Our method is in principle not limited to three colors and can be generalized e.g. for 4 color imaging using 2 spectral channels and third order SOFI spectral cross-cumulant analysis (see theory in SI).
- Cumulant analysis enables multicolor super-resolved imaging when blinking conditions for single molecule localization microscopy are hard or impossible to achieve. Our approach facilitates fluorophore selection and experimental realization (compared to (d)STORM).
- Simultaneous data acquisition saves time, as the reviewer mentioned (compared to sequential SOFI or sequential exchange PAINT/SMLM). As well, photobleaching is a limiting factor for sequential imaging as premature bleaching of the fluorophores that are imaged last must be avoided. In addition, the longer the image acquisition, the more sample movement (in the case of live cells) and the more sample drift can occur (in general, needs careful correction) which can be detrimental for e.g. colocalization analysis.

This referees' argumentations has been a perfect stimulation to expand the scope of our manuscript for live cell imaging (see additional section on live cell imaging at the end of the manuscript). We performed additional new experiments showing three color live-cell imaging. We

used the reversibly photoswitchable fluorescent proteins fusion construct that we have applied previously for SOFI (Vimentin-Dreiklang) and two live cell compatible small molecule stains to label mitochondria (Mitotracker Deep Red FM) and wheat germ agglutinin (WGA)-AbberiorFlip565. The appropriate fluorophore blinking was achieved using 488 nm light for imaging and off-switching of Dreiklang, with on-switching via thermal relaxation; mitotracker was imaged and switched off by 635 nm excitation and the self-blinking spiroamide dye conjugate WGA-AbberiorFlip565 was imaged using 561nm excitation. This proof-of-principle multicolor experiments shows the cytoskeleton elements Vimentin (blue), mitochondria (red) and WGA (presumably WGA accumulated in internalized vesicles or membrane folds as we are focusing just above the basal membrane, green). This can be seen in the three color unmixed images in Figure R1b) and Figure R2 below. Figure R1c) illustrates the multicolor workflow for a zoomed in region in our live cell example. The initial images in the reflection channel I_R contain mostly signal from the Vimentin-Dreiklang fluorescent protein construct mixed with sparse WGA agglomeration (see arrows in Figure R1c) for WGA overlaying Vimentin filaments). The spectral cross-cumulant image $\kappa_{2,RR}$ reflects the blinking signals and the spectral unmixing finally removes the WGA-AbberiorFlip565 signal from the Dreiklang channel $\tilde{\chi}_2\{I_1\}$ to uncover previously hidden Vimentin structure (see also the line profiles in Figure R1c) and note the background reduction). Figure R3 below shows the corresponding images for all channels.

Figure R1 Multicolor SOFI live cell imaging of the cytoskeleton, nucleus and wheat-germ agglutinin in COS-7 cells. a) Overlay of the average intensity acquired in the reflection (green) and transmission (red) channel of live cells in Hanks balanced salt solution and about 0.8 kWcm^{-2} 488 nm, 0.8 Wcm^{-2} 561 nm and $<1 \text{ kWcm}^{-2}$ 635 nm illumination intensity. Scale bar 5 μm . b) RGB composite image of the unmixed, flattened and deconvolved SOFI images with Mitotracker Deep Red FM stained mitochondria (red, $\tilde{\chi}_2\{I_3\}$), accumulated wheat germ agglutinin-AbberiorFlip565 labeling (green, $\tilde{\chi}_2\{I_2\}$) and Vimentin-Dreiklang fluorescent protein expression (blue, $\tilde{\chi}_2\{I_1\}$). Scale bar 5 μm . c) Close-up of the ROI indicated in a) and b) showing the mean intensity in the reflection channel I_R , the second order spectral cross-cumulant image $\kappa_{2,RR}$ and the unmixed image in the Dreiklang channel $\tilde{\chi}_2\{I_1\}$, all displayed using the morgenstemming colormap²⁷ and comparison of the normalized intensity profiles along the indicated line (green I_R , black $\kappa_{2,RR}$, blue $\tilde{\chi}_2\{I_1\}$). Scale bar 2 μm .

Figure R2 Multicolor SOFI live cell imaging of the cytoskeleton, nucleus and wheat-germ agglutinin in COS-7 cells. a) RGB composite image of the unmixed, flattened and deconvolved SOFI images with b) Mitotracker Deep Red FM stained mitochondria (red, $\tilde{\chi}_2\{I_3\}$), c) Vimentin-Dreiklang fluorescent protein expression (blue, $\tilde{\chi}_2\{I_1\}$) and d) accumulated wheat germ agglutinin-AbberiorFlip565 labeling (green, $\tilde{\chi}_2\{I_2\}$). Data from Figure R1. Scale bar 5 μm , all displayed using the morgenstemning colormap²⁷.

Figure R3 Multicolor SOFI live cell imaging; spectral cross-cumulants and unmixed, flattened and deconvolved SOFI a) RGB composite image of the second order spectral cross-cumulant images with $\kappa_{2,RR}$ (green), $\kappa_{2,RT}$ (blue) and $\kappa_{2,TT}$ (red) and single channel images of the ROIs indicated in Figure R1 a) and b). b) RGB composite image of the unmixed, flattened and deconvolved

*SOFI images with Mitotracker Deep Red FM stained mitochondria (brightness enhanced, red, $\tilde{\chi}_2\{I_3\}$), accumulated wheat germ agglutinin-AbberiorFlip565 labeling (green, $\tilde{\chi}_2\{I_2\}$) and Vimentin-Dreiklang fluorescent protein expression (blue, $\tilde{\chi}_2\{I_1\}$) and single channel images of the ROIs indicated in Figure R1 a) and b). Scale bars 2 μm . Single channel images are displayed using the *morgenstemning* colormap²⁷. Scale bars 2 μm .*

Reviewer #2:

Minor

1. In Figure 3, the authors could add some analysis to the performance of the method, rather than just the figure and the zoom.

Reply:

Unfortunately, analysis of remaining cross-talk after application of our method is not easily possible in the experiments due to overlapping structures. This is why we report these performance parameters for different simulated datasets (see manuscript and Supplementary Table 2 and 3). For the new live cell imaging data acquired for the revision, we added visualization of the cross-talk and unmixing in the Dreiklang channel (see Figure R1c)). We attempted to estimate the cross-talk for the WGA-AbberiorFlip565 channel for areas with non-overlapping structures, but found these difficult to identify and the analysis unreliable.

2. Figure 3 and 4 provided similar points of color unmixing and did not add much.

Reply: We agree that both Figures demonstrate successfully color unmixing. Our goal was to show that the method works for different cell types and for different intracellular targets including highly overlapping structures as presented in Figure 4. We replaced the previous Figure 4 by the new Figure on live cell imaging and moved Figure 4 into the SI.

Reviewers' Comments:

Reviewer #2:

Remarks to the Author:

In response to my previous comment, the authors made considerable efforts in improving the paper, mostly by adding the proof-of-principle live cell 3-color experiment. However, it is still hard to evaluate the performance of the technique without some quantitative measurement. What is the resolution in each color (experimental and theoretical)? What is the registration accuracy between different color? How long does it take to take an image and how does movement of fluorophore blur the super-resolution image? How does it compare with other super-resolution techniques and what are the unique advantages? As a method paper, I think it is important to give these characteristics for the reader to decide whether to use the method.

We are thankful to the reviewer for reevaluating our manuscript. We addressed the remarks point-by-point as described below.

All the changes, additional information and Figures in the revised version of the manuscript and supplementary information are highlighted in yellow.

Reviewer #2 (Remarks to the Author):

Reviewer #2: In response to my previous comment, the authors made considerable efforts in improving the paper, mostly by adding the proof-of-principle live cell 3-color experiment.

Reply: We thank the reviewer for the positive reception of the live cell imaging that we added to the manuscript.

Reviewer #2: However, it is still hard to evaluate the performance of the technique without some quantitative measurement. What is the resolution in each color (experimental and theoretical)? What is the registration accuracy between different color?

Reply:

We added a section on the resolution obtained with our multicolor approach in the supplementary information (see *Multicolor SOFI with spectral unmixing: additional data and resolution estimation*):

Our multicolor approach is based on cumulant analysis and we used the same theoretical framework originally devised for spatially super-resolved SOFI. The theoretically attainable resolution increase for second order analysis with respect to widefield imaging is two-fold after spectral cross-cumulation and post-processing (using deconvolution and linearization¹ (used here) or Fourier reweighing²).

We estimated the resolution of our imaging using image decorrelation analysis in ImageJ (default settings)³. This approach uses partial phase autocorrelation for a series of filtered images to determine the highest spatial frequency with sufficiently high signal in relation to noise. For the fixed cells in Figure 3 (microtubules-Alexa488 λ_{em} = 519 nm, WGA-Atto565 λ_{em} = 592 nm and Lamin B1-Alexa647 λ_{em} = 665 nm), the estimated resolutions are 317 nm in the reflected channel and 325nm in the transmitted channel (average image of the time series). Both channels contain signals from all three dyes, albeit at different proportions, and camera noise as well as out-of-focus signal.

The theoretical widefield resolution of ideal point sources according to the Abbe criterion $d = \frac{\lambda_{em}}{2NA}$ (NA=1.27) is estimated at the maximum fluorescence emission to 204 nm, 233 nm and 262 nm for the three dyes (disregarding the fluorescence emission tail at longer wavelengths); the corresponding theoretical SOFI resolution is twofold improved 102 nm, 117 nm and 131 nm. Taking into account the underlying structure of e.g. microtubule apparent diameter of 25 nm + 40 nm (secondary immunostaining increases the imaged microtubule diameter by 20-40 nm⁴), the expected theoretical resolution in the absence of noise e.g. in the Alexa488 unmixed SOFI channel is approximately $\sqrt{65^2 + 102^2} \text{ nm} = 121 \text{ nm}$.

After deconvolution and linearization, we estimate 151 nm in the Alexa488 channel (2.09 fold improvement vs. I_R), 154 nm in the Atto565 channel (2.06/2.11 fold improvement vs. $I_{R/T}$) and 159 nm in the Alexa647 channel (2.04 fold improvement vs. I_T) for the results shown in Figure 3. The estimations and fold improvement are reasonable considering the finite resolution assessment

accuracy and considering that the widefield images contain a mixture of the three fluorophores and background fluorescence.

We also added a discussion regarding the impact of the image coregistration (see *Co-registration of physical color channels*):

The coregistration procedure is a preprocessing step independent of our proposed approach for multicolor imaging. We generally perform coregistration of physical color channels using affine transformation and bilinear interpolation based on calibration images of corresponding beads that are visible in both channels (implemented in Matlab). This is a standard procedure used in many labs yielding subpixel precision⁵. For example, the coregistrations for our data has a precision (vector sum of coordinate displacement in original vs. coregistered channel) of ~10-30 nm. The virtual spectral cross-cumulant channel has by construction half the registration error with respect to the physical color channels. The unmixing step is a linear operation, thus the registration between the final unmixed color channels is comparable to the coregistration precision of the physical color channels - an order of magnitude better than the attainable resolution for second order analysis. It is noteworthy that careful coregistration is important for cross-cumulant analysis, but sufficient accuracy is routinely achieved^{6,7}.

Reviewer #2: How long does it take to take an image and how does movement of fluorophore blur the super-resolution image?

Reply: For imaging of fixed cells, we analyzed 2000 frames at 20 ms exposure time corresponding to 40 s acquisition. For live cell imaging, we analyzed 1000 frames at 20 ms exposure time corresponding to 20 s acquisition. We added these numbers to the figure legends (in addition to mentioning it in the text) and apologize that the information was missing for the live cell data.

We added the following paragraph on live cell super-resolution imaging remarks in the manuscript:

Live cell super-resolution imaging is challenging, considering the intricate relationship between phototoxicity, imaging speed, resolution increase, spatial sampling and possible motion blur⁸. For biological investigations, these parameters need to be carefully considered for each target and fluorophore. The directional movement of fluorophores during the imaging limits the useful spatio-temporal window where correlations can be extracted which in turn lowers the SOFI signal-to-noise ratio and the resolution. In general, the resolution is compromised when the frame acquisition time and the feature velocity lead to a displacement on the order of the attainable resolution⁹. Diffusion of fluorophores for an overall stationary structure has only minor effects on the SOFI signal while substantially improving the spatial sampling¹⁰. In general, SOFI tolerates higher labeling densities, on-time ratios, lower signal-to-noise ratio and needs less frames to reconstruct an image than SMLM (see also below). There have been select applications where single-color SMLM achieved < 1s time resolution, but usually tens of seconds acquisitions are necessary^{8,11}. We previously demonstrated live cell SOFI with acquisition times ~ 1s (~ 325 frames⁷) using optimized, fast switching of Dreiklang and others achieved live cell SOFI for several fluorescent proteins by analyzing ~500 frames¹².

The further optimization of the switching rates and minimization of acquisition times for multicolor SOFI with spectral unmixing is possible, but beyond the scope of this paper.

Reviewer #2: How does it compare with other super-resolution techniques and what are the unique advantages? As a method paper, I think it is important to give these characteristics for the reader to decide whether to use the method.

Reply: As a result of the referees comment, we decided to add a paragraph to the Summary and Discussion section that compares our technique with other super-resolution techniques and highlights the unique advantages. We would like to remark that, in addition to presenting a new SOFI-based multicolor imaging approach, our results also represent the first demonstrations of 3-color SOFI (3 different colors labeling three different structures) in fixed and living cells.

The new paragraph reads:

Spectral cross-cumulant multicolor SOFI exploits the crosstalk between adjacent spectral channels. The crosstalk in this concept is key in extracting the third color and is not a limitation or artefact like in classical multicolour imaging. This concept exploits the intrinsic properties of cumulant analysis in the spectral domain while maintaining the super-resolution performance in the spatial domain. Alternative super-resolution concepts based on structured illumination or targeted switching⁸ require an instrument modification adapted for each spectral channel. Super-resolution approaches relying on the stochastic fluorophore switching generally require simpler instrumentation. Our approach does not need any hardware modification and can be easily adopted in ubiquitous 2-channel widefield microscopes. Compared to other super-resolution multicolor approaches, we used standard labeling with organic dyes and fluorescent proteins compatible with live cell imaging. There is no need for sophisticated probes that are difficult to obtain and use (e.g. DNA probes for exchange-PAINT¹³ or activator-reporter pairs in STORM¹⁴). Our analysis preserves all the advantages inherent to SOFI such as optical sectioning, elimination of uncorrelated background and increased spatial resolution¹⁵. In particular, cumulant analysis tolerates high fluorophore labelling densities, high on-time ratios and low signal-to-noise ratios^{16,17}. This facilitates fluorophore selection and experimental realization (e.g. no need for high power lasers), which is especially important for multicolor imaging. Spectral cross-cumulation thus enables multicolor super-resolved imaging when blinking conditions for single molecule localization microscopy are hard or impossible to achieve^{16,18,19}. The demonstrated 2nd order analysis offers 3-color imaging with up to twofold resolution increase. In principle, our approach is not limited to three colors; e.g. 4-color imaging with 2 spectral channels could be achieved through third order SOFI at up to 3-fold higher resolution (see theory in the Supplementary Information). Obviously, simultaneous data acquisition saves time compared to sequential imaging and SOFI generally achieves super resolution using less frames and shorter acquisition times as well as lower laser intensities than SMLM^{16,17}. This is important for live-cell imaging to minimize motion blur and phototoxicity.

- (1) Geissbuehler, S.; Bocchio, N. L.; Dellagiacomma, C.; Berclaz, C.; Leutenegger, M.; Lasser, T. Mapping Molecular Statistics with Balanced Super-Resolution Optical Fluctuation Imaging (BSOFI). *Opt. Nanoscopy* **2012**, *1* (1), 1–7. <https://doi.org/10.1186/2192-2853-1-4>.
- (2) Dertinger, T.; Colyer, R.; Vogel, R.; Enderlein, J.; Weiss, S. Achieving Increased Resolution and More Pixels with Superresolution Optical Fluctuation Imaging (SOFI). *Opt. Express* **2010**, *18* (18), 18875. <https://doi.org/10.1364/oe.18.018875>.
- (3) Descloux, A.; Größmayer, K. S.; Radenovic, A. Parameter-Free Image Resolution Estimation Based on Decorrelation Analysis. *Nat. Methods* **2019**, *16* (9), 918–924. <https://doi.org/10.1038/s41592-019-0515-7>.

- (4) Mikhaylova, M.; Cloin, B. M. C.; Finan, K.; Van Den Berg, R.; Teeuw, J.; Kijanka, M. M.; Sokolowski, M.; Katrukha, E. A.; Maidorn, M.; Opazo, F.; Moutel, S.; Vantard, M.; Perez, F.; Van Bergen En Henegouwen, P. M. P.; Hoogenraad, C. C.; Ewers, H.; Kapitein, L. C. Resolving Bundled Microtubules Using Anti-Tubulin Nanobodies. *Nat. Commun.* **2015**, *6* (May). <https://doi.org/10.1038/ncomms8933>.
- (5) Deschout, H.; Shivanandan, A.; Annibale, P.; Scarselli, M.; Radenovic, A. Progress in Quantitative Single-Molecule Localization Microscopy. *Histochem. Cell Biol.* **2014**, *142* (1), 5–17. <https://doi.org/10.1007/s00418-014-1217-y>.
- (6) Descloux, A.; Grubmayer, K. S.; Bostan, E.; Lukes, T.; Bouwens, A.; Sharipov, A.; Geissbuehler, S.; Mahul-Mellier, A. L.; Lashuel, H. A.; Leutenegger, M.; Lasser, T. Combined Multi-Plane Phase Retrieval and Super-Resolution Optical Fluctuation Imaging for 4D Cell Microscopy. *Nat. Photonics* **2018**, *12* (3), 165–172. <https://doi.org/10.1038/s41566-018-0109-4>.
- (7) Geissbuehler, S.; Sharipov, A.; Godinat, A.; Bocchio, N. L.; Sandoz, P. A.; Huss, A.; Jensen, N. A.; Jakobs, S.; Enderlein, J.; Gisou Van Der Goot, F.; Dubikovskaya, E. A.; Lasser, T.; Leutenegger, M. Live-Cell Multiplane Three-Dimensional Super-Resolution Optical Fluctuation Imaging. *Nat. Commun.* **2014**, *5*, 1–7. <https://doi.org/10.1038/ncomms6830>.
- (8) Schermelleh, L.; Ferrand, A.; Huser, T.; Eggeling, C.; Sauer, M.; Biehlmaier, O.; Drummen, G. P. C. Super-Resolution Microscopy Demystified. *Nat. Cell Biol.* **2019**, *21* (1), 72–84. <https://doi.org/10.1038/s41556-018-0251-8>.
- (9) Shroff, H.; Galbraith, C. G.; Galbraith, J. A.; Betzig, E. Live-Cell Photoactivated Localization Microscopy of Nanoscale Adhesion Dynamics. *Nat. Methods* **2008**, *5* (5), 417–423. <https://doi.org/10.1038/nmeth.1202>.
- (10) Vandenberg, W.; Dedecker, P. Effect of Probe Diffusion on the SOFI Imaging Accuracy. *Sci. Rep.* **2017**, *7* (February), 1–8. <https://doi.org/10.1038/srep44665>.
- (11) Lakadamyali, M. Super-Resolution Microscopy: Going Live and Going Fast. *ChemPhysChem* **2014**, *15* (4), 630–636. <https://doi.org/10.1002/cphc.201300720>.
- (12) Duwé, S.; Moeyaert, B.; Dedecker, P. Diffraction-Unlimited Fluorescence Microscopy of Living Biological Samples Using PcSOFI. *Curr. Protoc. Chem. Biol.* **2015**, *7* (1), 27–41. <https://doi.org/10.1002/9780470559277.ch140025>.
- (13) Jungmann, R.; Avendaño, M. S.; Woehrstein, J. B.; Dai, M.; Shih, W. M.; Yin, P. Multiplexed 3D Cellular Super-Resolution Imaging with DNA-PAINT and Exchange-PAINT. *Nat. Methods* **2014**, *11* (3), 313–318. <https://doi.org/10.1038/nmeth.2835>.
- (14) Bates, M.; Dempsey, G. T.; Chen, K. H.; Zhuang, X. Multicolor Super-Resolution Fluorescence Imaging via Multi-Parameter Fluorophore Detection. *ChemPhysChem* **2012**, *13* (1), 99–107. <https://doi.org/10.1002/cphc.201100735>.
- (15) Dertinger, T.; Colyer, R.; Iyer, G.; Weiss, S.; Enderlein, J. Fast, Background-Free, 3D Super-Resolution Optical Fluctuation Imaging (SOFI). *Proc. Natl. Acad. Sci. U. S. A.* **2009**, *106* (52), 22287–22292. <https://doi.org/10.1073/pnas.0907866106>.
- (16) Geissbuehler, S.; Dellagiacoma, C.; Lasser, T. Comparison between SOFI and STORM. *Biomed. Opt. Express* **2011**, *2* (3), 408–420. <https://doi.org/10.1364/boe.2.000408>.

- (17) Deschout, H.; Lukes, T.; Sharipov, A.; Szlag, D.; Feletti, L.; Vandenberg, W.; Dedecker, P.; Hofkens, J.; Leutenegger, M.; Lasser, T.; Radenovic, A. Complementarity of PALM and SOFI for Super-Resolution Live-Cell Imaging of Focal Adhesions. *Nat. Commun.* **2016**, *7*, 1–11. <https://doi.org/10.1038/ncomms13693>.
- (18) Lukeš, T.; Glatzová, D.; Kvíčalová, Z.; Levet, F.; Benda, A.; Letschert, S.; Sauer, M.; Brdička, T.; Lasser, T.; Cebecauer, M. Quantifying Protein Densities on Cell Membranes Using Super-Resolution Optical Fluctuation Imaging. *Nat. Commun.* **2017**, *8* (1). <https://doi.org/10.1038/s41467-017-01857-x>.
- (19) Burgert, A.; Letschert, S.; Doose, S.; Sauer, M. Artifacts in Single-Molecule Localization Microscopy. *Histochem. Cell Biol.* **2015**, *144* (2), 123–131. <https://doi.org/10.1007/s00418-015-1340-4>.

Reviewers' Comments:

Reviewer #2:

Remarks to the Author:

The authors addressed my suggestions about the resolution and registration accuracy of the method.